# Role of Estrogen and Estrogen Receptor in GH-Secreting Adenomas

**DOI:** 10.3390/ijms24129920

**Published:** 2023-06-08

**Authors:** Giacomo Voltan, Pierluigi Mazzeo, Daniela Regazzo, Carla Scaroni, Filippo Ceccato

**Affiliations:** 1Department of Medicine (DIMED), University of Padova, Via Giustiniani 2, 35128 Padova, Italy; giacomo.voltan@aopd.veneto.it (G.V.); pierluigi.mazzeo@aopd.veneto.it (P.M.); daniela.regazzo@unipd.it (D.R.); carla.scaroni@unipd.it (C.S.); 2Endocrinology Unit, Padova University Hospital, Via Ospedale Civile 105, 35128 Padova, Italy

**Keywords:** acromegaly, SERMs, estrogen, IGF1, estrogen receptor

## Abstract

Acromegaly is a rare disease with several systemic complications that may lead to increased overall morbidity and mortality. Despite several available treatments, ranging from transsphenoidal resection of GH-producing adenomas to different medical therapies, complete hormonal control is not achieved in some cases. Some decades ago, estrogens were first used to treat acromegaly, resulting in a significant decrease in IGF1 levels. However, due to the consequent side effects of the high dose utilized, this treatment was later abandoned. The evidence that estrogens are able to blunt GH activity also derives from the evidence that women with GH deficiency taking oral estro-progestins pills need higher doses of GH replacement therapy. In recent years, the role of estrogens and Selective Estrogens Receptor Modulators (SERMs) in acromegaly treatment has been re-evaluated, especially considering poor control of the disease under first- and second-line medical treatment. In this review, we analyze the state of the art concerning the impact of estrogen and SERMs on the GH/IGF1 axis, focusing on molecular pathways and the possible implications for acromegaly treatment.

## 1. Somatotroph Adenomas and Acromegaly

Acromegaly is a rare, chronic, and systemic disease caused by excessive secretion of growth hormone (GH), which leads to increased circulating insulin-like growth factor 1 (IGF1). Recent studies reported a prevalence of >13 cases for every 100,000 individuals [1,2], whereas the estimated annual incidence is up to 1.1 cases/100,000 people [3,4]. A soft predominance of female acromegalic patients is reported ranging from 52% to 60% of the prevalence [5]. Interestingly, the diagnostic delay of the disease is 2–6 times longer in female patients, despite generally earlier consultations with physicians [6]. In more than 95% of cases, the etiology is a pituitary GH-secreting adenoma [7]. More rarely, an ectopic secretion of growth hormone-releasing hormone (GHRH) was reported, mainly sustained by pancreatic or pulmonary carcinoid tumors [8].

The pathogenesis of somatotroph adenomas is not fully understood; nonetheless, they are usually benign and sporadic, though several alterations in cell signaling have been described. GH-secreting pituitary adenomas arise as monoclonal expansions of well-differentiated somatotroph cells, derived from the transcription factor PIT1 that drives the lineage of mammosomatotroph differentiation [9]. The PIT1-lineage adenomas are classified as pure GH adenomas in the 2022 WHO classification, and they can be further classified as densely or sparsely granulated. The activating mutations in the *GNAS1* gene, usually a substitution of Arg201 or Gln227 residues, occur in 40% of GH-secreting adenomas, causing a constitutive and uncontrolled production of GH due to the accumulation of cAMP. These patients show a peculiar phenotype: they are usually older, with smaller and less invasive densely granulated tumours [10]. Considering the non-genomic factors contributing to adenoma pathogenesis, the signal transducer and activator of transcription 3 (STAT3) is an overexpressed gene in somatotroph adenomas [10]. Genomic profiling of GH-secreting pituitary adenomas has revealed chromosome copy number alterations in 30% of somatotroph adenomas, higher than those of other secreting or not-functioning adenomas [9,10].

Moreover, in most patients with a paradoxical increase of GH during an Oral Glucose Tolerance Test (OGTT), an overexpression of the glucose-dependent insulinotropic polypeptide (GIP) receptor was found in somatotroph adenomas [11]. Furthermore, acromegaly may be associated with various genetic syndromes such as MEN1, Carney-Complex, McCune-Albright, FIPA, MEN4 or X-LAG [12,13].

After clinical suspicion, the diagnosis of acromegaly is based upon endocrine features (elevation IGF-1, unsuppressed GH during OGTT) and radiological confirmation of the adenoma in a pituitary magnetic resonance [10]. Several comorbidities are related to chronic GH and/or IGF1 excess, such as cardiovascular, metabolic, osteoarticular and neoplastic complications, which increase the overall mortality of uncontrolled acromegaly [10]. Notably, the prevalence of hypertension and diabetes seems to be greater in female acromegalic patients than in male ones [6].

Therefore, the main aims of treatment are to normalize GH and IGF1 levels, to reduce clinical symptoms and to decrease the rate of morbidity and mortality [14,15]. Today, the treatment of acromegaly should be personalized to patient characteristics and managed by a multidisciplinary team, operating in a referral center [16,17,18]. Transsphenoidal resection of the somatotroph adenoma is still the first-line treatment recommended by current guidelines [14,15]; however, the remission rate is between 40–60% after surgery, higher in microadenomas than in macroadenomas [19].

Consequently, medical therapy plays an important role, either as a first-line treatment in patients not eligible for surgery or as a second-line treatment. Among the availa-ble drugs, injectable first-generation somatostatin receptor ligands (SRLs) Lanreotide and Octreotide are recommended as initial treatment [14,15,20]. Pasireotide (the second-generation SRL) and Pegvisomant (the GH receptor antagonist) are usually indicated as second-line treatments, whereas Cabergoline (CAB) may be a reasonable choice in the case of mild disease or in addition to SRLs [21]. Either first- or second-generation SRLs are able to induce significant tumor shrinkage [22,23]. The true efficacy of medical therapy is well known, and it does not control hormonal excess in all patients with acromegaly: control rates of first-generation SRLs treatment were 56% for mean GH and 55% for IGF1 normalization [24]; the rate of biochemical control of acromegaly during Pasireotide ranged from 27% to 93% of cases [25]; finally, several studies have shown that Pegvisomant is efficient in normalizing IGF1 hypersecretion from 58% to 97% of patients [26]. Therefore, the disease remains poorly controlled in a significant portion of acromegalic patients, even in cases of high-intensity schemes or combined treatment [27]. Moreover, drugs like Pasireotide or Pegvisomant are highly expensive, especially considering the need for ongoing treatments [28]. Interestingly, women have been shown to require higher doses of Pegvisomant to achieve an equivalent response to that in men in terms of IGF1 normalization, whereas treatment outcomes with SRLs seem to be comparable between both genders [6].

As of several years ago, estrogens have been shown to be effective in lowering IGF1 levels and controlling clinical symptoms in acromegalic patients [29]. In addition, the evidence that estrogen negatively regulates GH action is found in the need for a higher dose of GH replacement therapy in women taking contraceptive estro-progestin drugs [30]. Similar results in terms of efficacy in lowering IGF1 were reported using selective estrogen receptor modulators (SERMs), which mimic the effects of estrogen in some tissues while acting as anti-estrogen in other organs [31,32,33,34].

The aim of this review is to map the current literature regarding the state of the art regarding the impact of estrogens and SERMs on the GH/IGF1 axis, focusing on molecular pathways and the possible effects on acromegaly treatment. To the best of our knowledge, the only attempt to collect the available evidence was a mini-review published by Duarte et al. in 2016 [35].

The MEDLINE (PubMed) database was queried, from database inception to present (last search 15 March 2023), using a combination of Medical Subject Headings (MeSH) and free-text terms: (Acromegaly OR acromegaly resistance OR acromegaly treatment) AND (Estrogens OR SERMs). No further limits or filters were applied. Duplicate records were removed. Additional articles were identified with manual searches, including a thorough review of other review articles and relevant references. Titles, abstracts and keywords of retrieved records were screened for relevance. Furthermore, full texts were read by the authors.

## 2. The GH-IGF1 Axis and Estrogens: How They Work

GH is secreted by the somatotroph cells located in the anterior portion of the pituitary gland. GH secretion is pulsatile and regulated by several elements, exhibiting a stimulatory or inhibitory effect. The main stimulating factors are GH-releasing hormone, Ghrelin, estrogens, adrenergic peptides, amino acids and some conditions like hypoglycemia, deep sleeping and stress, as depicted in Figure 1 [36,37].

Somatostatin (SST) is the most powerful GH-inhibiting factor. However, glucocorticoids, IGF1, glucose, hypothyroidism and obesity are also able to blunt GH secretion [38]. GH exerts pleiotropic effects, acting mainly in an indirect manner through the action of IGF1. However, GH is also responsible for a direct effect on chondrocytes, promoting their growth and proliferation, and on glycemic and lipid metabolism [39].

GH elicits intracellular signaling through specific GH-receptors (GHRs) (Figure 2), which are transmembrane proteins that dimerize after GH binding and induce activation and phosphorylation of Janus kinase 2 (JAK2) [40]. JAK2 then phosphorylates GHRs and the signal transducer activators of transcription (STATs) 1–3 and 5 [41,42]. STATs themselves dimerize and translocate to the nucleus, where they promote the transcription of target genes, including IGF1 [43].

Some of these genes also encode for suppressors of cytokine signaling (SOCS), family proteins that are involved in the suppression of the JAK/STAT pathway [44]. Indeed, SOCS1-2 and -3 contain a Src Homology 2 domain through which they can bind and inhibit the catalytic domain of JAK2. The result is a feedback inhibition of GH-related transcriptional activity [44,45].

Estrogens perform their physiological actions through the estrogen receptor (ER), which is a member of the nuclear receptors (NRs) superfamily. NRs are transcriptional regulators containing a specific DNA-binding domain and a ligand-binding domain, which is usually highly conserved among superfamily members [46]. Two main isoforms of ER are presently known: ERα and ERβ. Both receptors, such as NR, have a ligand domain that binds endogenous and synthetic estrogens. ERα and ERβ are products of different genes and exhibit specific tissue and cell-type expressions. Furthermore, the DNA estrogen-responsive elements (ERE) vary between the two receptors [47]. It was observed that while ERα mediates the proliferative response to estrogen, ERβ decreases cell proliferation [48,49]. High levels of ERα are expressed in the hypothalamus and in the pituitary gland, whereas ERβ expression has also recently been revealed in the somatotroph [50].

## 3. Effects of Estrogens on GH-IGF1 Axis: What In Vitro and In Vivo Models Show Us

Estrogens regulate GH-IGF1 axis activity in several ways, affecting both pituitary GH secretion and peripheral hepatic IGF1 production.

Estrogens play a secretagogue role in GH secretion, either at the pituitary or hypo-thalamus. At first, they negatively modulate SST receptors (SSTRs) expression [51], reducing the somatostatinergic tone, which results in turn in enhanced GH secretion. Moreover, estradiol (E2) increases GHRH and decreases SST release in animal models [52], through direct control of GH synthesis at a pre-translational level: both ERα and ERβ act in the transcriptional control of GH in the somatotroph cell [50].

In support of this evidence, male mice have lower GH values, less random GH secre-tory bursts and longer periods between GH pulses than female mice. In SST knock-out models, male mice exhibited a feminized pattern of GH secretion, confirming the interplay between estrogens, SST and GH [53].

Estrogens also modulate the factors that regulate peripheral GH sensitivity (most of the estrogens’ modulation of the GH/IGF1 axis is presented in Table 1). As an example, ghrelin, a 28-amino-acid octanoylated stomach-secreted peptide [54], is one of the most potent endogenous GH secretagogues discovered: transdermal (E2) in healthy post-menopausal women augments hypothalamus-pituitary sensitivity to acylated ghrelin (with respect to those women randomized not to receive E2) [55].

In humans, baseline GH levels are higher in women when fasting and at rest than in men; conversely, men secrete more GH in response to SST-induced rebound [56]. Rather than systemic steroids, a paracrine stimulation of GH secretion seems to derive from local estrogens produced by intra-pituitary aromatization of testosterone in somatotroph cells [57]. In aromatase-knockout mice, E2 production was blocked, and the secretion of GH was low with elevated expression of SSTRs: E2 replacement therapy increased GH mRNA and reduced SST expression [58].

Contrary to the aforementioned positive effects of estrogens at the hypothalamus and pituitary level, there is similarly strong evidence that estrogens inhibit the GH receptor (GHR) and GH intracellular signaling that regulates IGF1 production in a dose-dependent manner [59]. Estrogens cause a reduction of IGF1 levels through three known mechanisms. Firstly, they lower the expression of GH receptors on cells, as seen in a study where exogenous administration of estradiol in rabbit liver cells induced a reduction in GHRs expression, measured both by GH binding and GH receptor mRNA levels [60]. Secondly, estrogen also suppresses GHR signaling through desensitization of the JAK/STAT pathway, mediated by phospholipase C (PLC) activation [61]. Indeed, in vitro, PLC binds to JAK2 with tyrosine phosphatase-1B, forming a ternary complex that reduces GH-induced JAK2 phosphorylation [62]. A summary of the studies reported in the literature is depicted in Figure 3.

This negative regulation decreases STATs phosphorylation and their nuclear translocation [61,62]. The final result is a blunted transcription of GH target genes, hampering, as a consequence, IGF1 secretion.

Thirdly, estrogens act by reducing JAK2 phosphorylation in an alternative way. K. C. Leung et al. demonstrated that, in vitro, this effect is mediated by an up-regulation of SOCS2 expression. As previously described, this protein in turn causes an impairment in JAK/STAT phosphorylation [59]. Later, these data were also confirmed in vivo on mice models, where it was shown that SOCS3 expression was augmented in the liver [63,64]. This evidence seems to confirm how the IGF1-lowering effect of estrogens and SERMS could mainly be the consequence of a peripheral action, not a direct inhibition of GH secretion, as already hypothesized [65]. Therefore, the fall in IGF1 concentration after oral estrogen therapy reduces the negative feedback on pituitary somatotroph cells, which could contribute to the indirect stimulation of GH secretion [36], previously described as a central estrogen-mediated effect.

However, in the literature there are some contrasting data. In immature lamb pituitary cells in vitro, tamoxifen induced a direct drop of GH release in somatotroph cells in a dose-dependent manner [66]. Contrary to this, Tulipano et al. demonstrated in human and rodent pituitary cells in vitro that a raloxifene analog (LY117018) could stimulate GH secretion through direct action [67], even if these results were not confirmed in vivo [13]. These contrasting data could be related both to distinct affinities for ER of the two molecules and to different ER expressions in those types of cells.

In fact, estrogen activity is mediated by activating two ER isoforms: ERα and Erβ. Therefore, the contrasting data found in vitro described above could be explained by the different binding affinities of natural and synthetic estrogens versus the two ER isoforms [68]. The IGF1 gene is mainly activated by anti-estrogens such as raloxifene or raloxifene-like molecules, but not by tamoxifen. That activation is specifically mediated by ERα [69].

Estrogens are also thought to play a pivotal role in testosterone-mediated GH production stimulation by the somatotroph cells. Indeed, testosterone replacement therapy in men with hypogonadism is associated with increased GH secretion [70]. However, the same effects were not reported in testing non-aromatizable androgens (dihydrotestosterone and dehydroepiandrosterone in humans) [71]. In addition, simultaneous treatment with aromatase inhibitors attenuates testosterone-induced GH stimulation [72]. Therefore, the testosterone conversion to estradiol seems to be fundamental in mediating GH production. The role of progesterone versus the GH/IGF1 axis is poorly studied; however, a recent paper reported that progesterone administration to post-menopausal women is associated with a blunted GH secretion after stimulation with ghrelin [73].

## 4. Clinical Use of Estrogens and/or SERMs in Patients with Acromegaly

### 4.1. The Use of Oral Estrogens in Acromegaly

Considering the IGF1-lowering effect of estrogen and SERMs, increasing attention has been paid over the last few decades to their application as a possible treatment in acromegalic patients. The use of estrogen and/or SERMs in acromegalic patients is highlighted in Table 2. Estrogens were first used in acromegaly over 40 years ago, improving clinical symptoms and glucose metabolism and decreasing IGF1 by almost 50% of the starting values [74,75]. However, this type of treatment was later abandoned, in light of the relevant side effects of the high dosage of ethinylestradiol administered, ranging from 0.5 to 1 mg. These effects were mainly related not only to the occurrence of nausea and weight gain [74], but also to the increased risk of new-onset thromboembolic events, coronary heart disease and breast cancer, as later reported in the WHI trial [76]. Moreover, estrogen treatment was not applicable in men because of the development of hypogonadism and gynecomastia.

After the advent of low-dose estro-progestin pills, physicians paid new attention to their use to treat acromegaly, alone or as an add-on to standard therapies. In a report by Cozzi et al., eight acromegalic women (aged 30–52 years) underwent a triphasic combined oral contraceptive (COC) treatment containing ethinylestradiol (30–40–30 mcg/day) and desogestrel (50–70–100 mcg/day). The treatment was conducted for a mean time of six months. Two patients were on combined therapy of Octreotide-LAR plus Cabergoline, three were receiving Octreotide-LAR and three patients were not taking any drugs. They observed a significant decrease in mean serum IGF1 by 45% in six patients (three with no therapy, two on SRLs, one on SRL + Cabergoline), four of which achieved normalization of IGF1. Interestingly, GH levels did not change, whereas in two patients, serum IGF1 sharply increased [77].

Vallette et al. in 2010 treated 11 acromegalic women with a 20 mcg ethinylestradi-ol/100 mcg levonorgestrel COC. Seven patients received COC plus Octreotide-LAR, while four patients received COC alone for a mean duration of 3.1 years. A 56.8% reduction of IGF1 levels was reported, leading to complete hormonal control in 73% (8/11) of patients; among them 91% (7/8) were treated with the combined therapy. No changes in GH concentration were observed [78].

In 2012, Shimon and Barkan published their experience with four acromegalic wom-en, three receiving COC, one receiving COC alone, one receiving COC plus Pegvisomant and one receiving COC plus Octreotide-LAR. All three patients achieved hormonal control, with IGF1 reduction ranging from 34% to 68% [79]. Interestingly, the woman treated with COC+pegvisomant was previously uncontrolled using COC+Octreotide-LAR and during seven days without taking COC, her IGF1 levels sharply rose, confirming the increasing effect of estrogen in IGF1 secretion. In these papers, IGF1 levels were assessed twice per year, providing a faithful kinetics of IGF1 changes during COC treatment. Transdermal estrogens were prescribed instead to the fourth patient in association with Octreotide-LAR, showing a considerable drop in IGF1 levels, albeit complete hormonal control was not achieved. As previously mentioned, the influence of estrogen on the GH-IGF1 axis depends on the route of administration. Hence, transdermal administration is not thought to reduce circulating IGF1 due to the lack of a hepatic first-pass effect. [80].

However, other authors demonstrated that estrogens also given through transdermal administration are able to suppress serum IGF1 concentrations, exactly as with oral administration. However, this is true only provided that sufficient estradiol levels are achieved in peripheral and hepatic blood [81]. Therefore, this might explain the IGF1 reduction response after transdermal estrogen reported by Shimon and Barkan [79].

Recently, a Brazilian group [82] assessed the efficacy of COC therapy containing ethinylestradiol 0.03 mg and levonorgestrel 0.15 mg in a group of eight acromegalic sub-jects previously treated with unsuccessful transsphenoidal surgery. After six months of COC therapy, 37% of patients (3/8) normalized IGF1 levels, two of whom received SRLs treatment, while in 25% of cases (2/8) a partial response was observed. Two patients did not respond to COC, showing a 13% increase in IGF1 levels in one case, while hormonal levels did not change in the last case. Moreover, mean GH values rose in two responsive and non-responsive subjects. In particular, the authors described a lack of expression of ER-α in patients who responded to oral estrogen therapy in terms of IGF1 normalization. Conversely, the presence of ERα was found in a patient who did not respond to estrogen treatment, and this was the only patient whose tumor grew during therapy. Hence, it might be speculated that ER-α expression could be a negative prognostic factor for the use of estrogens in somatotropinomas.

Interestingly, in human somatotropinomas, ERα mRNA expression usually did not occur in pure GH-tumors [83,84,85], while ERβ was detected in the majority of tumors secreting GH with mixed pathology [49]. Moreover, the findings of a different ER expression in pituitary adenomas could also generate the development of a highly specific ERβ agonist to regulate GH secretion in somatotropinomas, as recently discovered for the treatment of ovarian cancer [86,87].

In prolactinomas, tumor growth related to oral estrogen therapy has already been re-ported, since these types of tumors present a significant ER expression [88,89]. An important role might also be played, especially in males, by the intra-tumoral activity of aromatase, which locally transforms testosterone into estradiol [90]. Recently, we used anastrozole (an aromatase-inhibitor) to treat four male patients harboring macro-prolactinomas resistant to Cabergoline: a reduction of prolactin levels and a significant shrinkage of the adenomas was observed in all cases [91]. However, despite the fact that aromatase expression was proved in GH-secreting pituitary cells [57], a therapeutic approach with aromatase inhibitors has never been investigated in acromegalic patients.

### 4.2. Targeting the Estrogen Receptor with SERMs in Acromegaly

SERMs are synthetic drugs that bind estrogen receptors, exerting an agonist or antagonistic action depending on the different tissues. Usually, they block estrogen effects in the central nervous system (brain and pituitary gland) and breast, instead enhancing estrogenic effects in the cardiovascular system, bone and liver [92]. These drugs have been largely used in the adjuvant treatment of ER-positive breast cancer, showing a significant increase in overall survival [93]. There are few differences among the various SERMs, since it was reported that raloxifene induced a lower decrease in IGF1 levels than tamoxifen, considering both drugs were administered at a maximum dosage of 120 mg/day and 20 mg/day, respectively [94].

**Table 2 ijms-24-09920-t002:** The clinical use of estrogen and/or SERMs in acromegalic patients. The number of the reference (the same that appears in the manuscript) is depicted in the first column.

Studies	N of Patients	Male/ Female	Drug Used	Concomitant Therapy	GH Effect	IGF1 Relative Reduction	IGF1 Normalization
			Oral Estrogen				
Cozzi et al. (2003) [77]	8	0/8	Ethinylestradiol 30–40–30 mcg/day + desogestrel 50–70–100 mcg/day	OCT + CAB (2/8), OCT (3/8)	=	45% in 6/8 patients	4/8 (50%)
Vallette et al. (2010) [78]	11	0/11	Ethinylestradiol 20 mcg + levonorgestrel 100 mcg	OCT (7/11)	=	56.8%	8/11 (73%)
Shimon and Barkan (2012) [79]	4	0/4	ethinylestradiol 20 mcg + Gestodene 75 μg or ethinyl-estradiol 0.035 mg + cyproterone acetate 2 mg or Transdermal estrogen	PEG (1/4), OCT (2/4)	=	34 to 68%	3/4 (75%)
Magalhães et al. (2022) [82]	8	0/8	ethinylestradiol 0.03 mg and levonorgestrel 0.15 mg	OCT or LAN (6/8)	=/↑	21 to 54%	3/8 (37%)
			*SERMs*				
Cozzi et al. (1997) [95]	19	6/13	Tamoxifen 40 mg/die	none	↑	18% to 60%	4/19 (21%)
Balili et al. (2014) [31]	17	15/2	Tamoxifen 20–40 mg/die	OCT + PEG (1/17) or CAB (1/17) or OCT alone (1/17)	=	17.5%	8/17 (47%)
Mirfakhraee et al. (2021) [96]	1	0/1	Tamoxifen	anastrazole	↑	60%	1/1 (100%)
Attanasio et al. (2003) [97]	13	0/13	Raloxifene 60 mg/die	OCT (3/13), CAB (1/13)	=	35%	7/13 (54%)
Dimaraki et al. (2004) [34]	8	8/0	Raloxifene 60 mg twice a day	OCT (2/8)	=	16%	2/8 (25%)
Duarte et al. (2016) [35]	16	16/0	Clomiphene citrate 50 mg/die	OCT alone (4/16), OCT + CAB (7/16), CAB alone (5/16)	=	41%	7/16 (44%)
Koroglu et al. (2022) [98]	1	1/0	Clomiphene citrate 25 mg/die	LAN	No data	51%	1/1 (100%)

↑: Increased effect, =: neutral effect, GH: growth hormone, IGF1: insulin-like growth factor 1, OCT: Octreotide; LAN: Lanreotide; CAB: Cabergoline; PEG: Pegvisomant.

Cozzi et al. [95] first tried to use tamoxifen as a possible treatment for acromegaly; in 1997, they treated 19 acromegalic subjects (6 males, 13 females) for two months with an in-creasing dosage, reaching 40 mg/day. The mean IGF1 decreased by 29.5%, ranging from 18% to 60%, in 13 of 19 patients, achieving complete hormonal control in four of them (21%). GH levels slightly increased versus baseline, whereas after tamoxifen withdrawal serum IGF1 promptly rose.

Many years later, Balili et al. [31] reported that 17 patients (15 males and 2 females) with resistant acromegaly were treated with tamoxifen (maximum dose 40 mg/day) for a median period of four months. A significant reduction of IGF1 was highlighted in 82% of patients, reaching disease control in 47% of cases. Serum IGF1 levels were reduced by 17.5%, while GH levels did not change significantly. Interestingly, they described an increase of testosterone levels in all eight males with available data, as a consequence of anti-estrogenic action on gonadotroph pituitary cells. No adverse effects were reported.

A recent case report [96] describes a 57-year-old woman with active acromegaly and concomitant triple-positive breast carcinoma. She was treated with tamoxifen as adjuvant therapy showing a complete normalization of IGF1 levels for over three years; in addition, random serum GH increased versus baseline values during treatment. In support of the efficacy of tamoxifen, it should be pointed out that during a three-month withdrawal period, IGF1 levels sharply increased.

Similarly, other SERMs have also been explored as a possible treatment for acromegaly. Attanasio et al. [97] analyzed the impact of raloxifene 60 mg/day in 13 post-menopausal women for a median period of six months. Nine were resistant to standard treatments, which were consequently suspended, while the remaining four subjects went on with SRLs therapy. Hormonal control was achieved in 54% of patients (7/13), whereas the mean reduction of IGF1 was 35%; serum GH levels did not change during the whole treatment. However, in two patients, raloxifene was not effective in lowering IGF1, even showing an increase in one of them. In the same way, raloxifene was later tested in eight male acromegalic patients [34], observing a slight decrease of 16% in IGF1, reaching complete disease control only in 25% of cases (2/8). In this study, GH levels remain unchanged.

Duarte et al. [35] in 2016 studied 16 males with uncontrolled acromegaly, demonstrating the efficacy of Clomiphene citrate (CC) as an add-on therapy to SRLs or Cabergoline. Patients were treated for three months with CC 50 mg/day, showing a mean reduction of IGF1 levels by 41% (ranging from 16.8% to 68.3%), which lead 44% of patients to achieve hormonal control. No changes in GH levels were reported, while testosterone levels rose in 10 patients, as previously reported for tamoxifen [31]. Moreover, a 40-year-old man with uncontrolled acromegaly, despite maximal SRLs treatment and concurrent hypogonadism, was recently treated by adding CC 25 mg/day to standard therapy. After three months, the authors reported a complete normalization of IGF1 and testosterone levels, lasting two years [98].

As discussed above, estrogens and SERMs have largely demonstrated significant IGF1-lowering activity (the main SERMs and other drugs acting on estrogen pathways are depicted in Table 3).

The role of estrogens and SERMs in acromegaly was assessed by a meta-analysis including six different studies, for a total of 63 patients (49 females and 14 males) [32]. Twenty-three patients received estrogen while 40 were instead treated with SERMs (21 Raloxifene, 19 Tamoxifen), showing a complete normalization of IGF1 levels with a rate ranging from 21% to 75% of subjects. Interestingly, the greatest efficacy in hormonal reduction was reported among women treated with estrogens, followed by women on SERMs, and finally men on SERMs. These findings may suggest that SERMs have a weaker effect in inhibiting IGF1 compared with estrogens, probably due to their mixed agonistic and antagonistic action on ER. Moreover, the lack of decrease in GH values during SERMs treatment confirms the absence of pituitary action in the context of acromegaly treatment, in contrast to other results described in vitro [66].

## 5. GH-IGF1 Axis and Tumor Development: The Role of Estrogens

GH and IGF1 promote cell proliferation, with a potential contribution to carcinogenesis and tumor progression [99,100,101]. As discussed above, it is clear that estrogens and SERMs are able to strongly decrease circulating levels of IGF1. Therefore, considering the potential permissive role of high GH/IGF1 levels, estrogen therapy might contribute to decreasing the risk of cancer development not only in conditions like acromegaly, characterized by an important excess of serum IGF1, but also in other tumors, whose development is linked to normal-high levels of GH/IGF1.

In acromegaly, the higher risk of developing colonic polyps with a consequent increased rate of colon neoplasia is well-known [102,103]. Although in a large retrospective cohort study a greater mortality from malignant disease and colon cancer in patients with elevated GH has been reported [104], overall cancer risk was slightly, but not significantly, increased in a recent 20-year cohort matched study [105].

Despite the presence of conflicting data regarding the incidence of cancer in acromegalic patients [106,107,108,109], screening colonoscopy and neck ultrasound imaging are recommended in these patients, because of the increased risk of colon polyps and thyroid nodules development [15].

It is important to keep in mind that acromegaly is a very rare disease, and much evidence about the potential association between GH-IGF1 axis hyperfunction and cancer comes from animal models and non-acromegalic patients with the same type of neoplasia thought to be more frequent in acromegalic patients. In the early 2000s, many papers were published showing a possible association in the Western population between circulating IGF1 and IGFBP3 levels and the risk of breast, prostate, lung and colorectal cancers [110,111,112,113].

**Table 3 ijms-24-09920-t003:** Main SERMs, SERDs, AI and estrogens with related molecular and pharmacological features. The references used in this table are [92,114,115,116,117].

Drug Name	Molecular Features	Drug Action	Side Effects	Indication
SERMs				
Tamoxifene	Mixed agonist/antagonist action on ER	Breast, brain → − Bone, endometrium, cardiovascular → + Vagina → +/−	Endometrial cancer, VTE, hot flushes, atrophic vaginitis	Adjuvant treatment in ER+ breast cancer
Raloxifene		Brest, brain → − Cardiovascular, bone → + Endometrium, vagina → =	VTE, hot flushes, leg cramps	Prevention and treatment of post-menopausal osteoporosis
Clomiphene		Breast, brain, endometrium, vagina → −	Headache, hot flushes, GI disturbance, ovarian enlargement	Treatment of ovulatory dysfunction in infertile women
Toremifene		Breast, brain → − Cardiovascular, bone → + Endometrium, vagina → +/−	VTE, hot flushes, atrophic vaginitis	Treatment of metastatic ER+ breast cancer
SERDs				
Fulvestrant	Pure ER antagonism	ER degradation, no estrogenic activity	Hot flushes, GI disturbances	Treatment of locally advanced/metastatic ER+ breast cancer
AI				
Anastrozole	Nonsteroidal	Reversible aromatase inhibition	Bone loss, nausea, hot flushes	Adjuvant treatment in ER+ breast cancer
Letrozole	Nonsteroidal	Reversible aromatase inhibition	Bone loss, nausea, hot flushes	Adjuvant treatment in ER+ breast cancer
Exemestane	Steroidal	Irreversible aromatase inhibition	Bone loss, Hypertension, atrophic vaginitis	Adjuvant treatment in ER+ breast cancer
Estrogens				
EE	Synthetic estrogen	Strong ER binding	VTE, breast cancer, endometrial cancer, ovarian cancer	Contraception, HRT
E2V	Natural Estrogen	Weak ER binding		

SERMs: Selective Estrogen Receptor Modulators, SERDs: Selective Estrogen Receptor Degraders, AI: Aromatase Inhibitors, EE: Ethinylestradiol, E2V: Valerate Estradiol, ER: Estrogen Receptor, −: antagonist action, +: agonist action, =: neutral action, +/−: weak antagonist action, VTE: Venous ThromboEmbolism, GI: GastroIntestinal.

In human mammary glands, the majority of the GH-IGF1 axis’ effects is mediated by serum and locally produced IGF1. However, in breast cancer, GH has been shown to enhance proliferation, survival, invasion and angiogenesis in cancer cells, independent of IGF1 [114,115]. The IGF signaling involves a complex system composed of three ligands (IGF1, IGF2 and insulin), many membrane receptors and six high-affinity binding proteins (IGFBP1 to IGFBP6) [116]. IGF1 and IGF2 could influence tumor growth and metastatic process through their effect on cell proliferation, angiogenesis and the epithelial-to-mesenchymal transition (EMT). Indeed, IGF1 and IGF2 are able to increase the production of hypoxia-inducible factor α [117], which is effectively a main stimulator of the expression of vascular endothelial growth factor [118].

In addition, GH can favor the suppression of epithelial markers like E-cadherin, up-regulating instead mesenchymal markers such as N-cadherin and vimentin [119]. These processes are fundamental steps in EMT. Besides all these negative effects, there is also some evidence regarding a contrasting action of IGFBP3 which may be protective against tumor progression through an anti-proliferative and pro-apoptotic effect [120].

Both endogenous and exogenous GH seems to stimulate cellular motility and invasiveness of prostate cancer cells, increasing metastatic potential [121,122]. As with breast cancer, IGFBP3 might contribute to the suppression tumor growth [123]. In addition, prostate-specific antigen, which is commonly used as a marker of prostate cancer, performs a protease activity that cleaves IGFBP3 [124]. Interestingly, IGF1 is related, in a ligand-independent manner, to the activation of the androgen receptor, which might be implicated in cancer progression [125,126].

In the setting of colon cancer, GH expression is strongly related to tumor size and lymph node metastasis. Furthermore, GH is able to enhance the oncogenicity and EMT in cancer stem cells [127]. Recent findings suggest that high GH-circulating levels might suppress the activity of target genes like p53, APC and PTEN, promoting neoplastic colon growth [128]. On the other hand, a potential anti-tumoral effect of IGFBP3 was proposed by Belizon et al., in light of evidence of an increased rate of colon cancer in mice with low IGFBP3 [129].

A possible influence of the GH axis was also demonstrated for lung cancer. A single-nucleotide polymorphism, resulting the in the amino acid change p.P495T, located in the GHR, has been strongly linked to higher lung cancer risk in white, Chinese and Afro-American women [130,131,132]. Recently, it was shown that the p.P495T variant is associated with an impairment of SOCS-2 activity, which actually traduces in a prolonged GH signal stimulation [133].

Interestingly, raloxifene was shown in some studies [134,135] to increase circulating IGFBP3, which is thought to play a protective role against malignancies. However, a recent meta-analysis did not confirm significant changes in IGFBP3 levels during raloxifene administration [136]. Furthermore, the well-known protective role of tamoxifene in breast cancer might also involve a drop in tissue IGF1 through the inhibition of ER. Considering the aforementioned evidence, it might be speculated that the administration of estrogens, for example in women seeking for hormonal contraception, could carry a potential anti-tumoral effect by decreasing IGF1, even in non-acromegalic patients.

## 6. Conclusions

Estrogens and SERMs significantly reduce IGF1 production, acting mainly through an impairment in GH signaling at a peripheral level. For these reasons, their use in patients with persistent acromegaly after surgery or unfeasible surgery could be a viable option. In addition, adopting SERMs and estrogens as an add-on therapy to standard treatment for acromegaly has shown to be effective in achieving disease control. Oral contraceptives might be the best choice for women at reproductive age, provided the absence of contraindications. On the other hand, SERMs could be more suitable for post-menopausal women, but especially for men, in which estrogens are inapplicable, considering their high efficacy in improving eventual hypogonadism. Moreover, acromegalic patients are a high-risk population for new-onset cancer. Therefore, an additional treatment that can control both diseases can be an optimal choice for patients.

Lastly, the modulation that estrogens/SERMs can induce on local GH and/or IGF1 excess or activity can be of interest in some types of high-impact cancers, studied in patients with acromegaly as a model and translated to the general population.

## Figures and Tables

**Figure 1 ijms-24-09920-f001:**
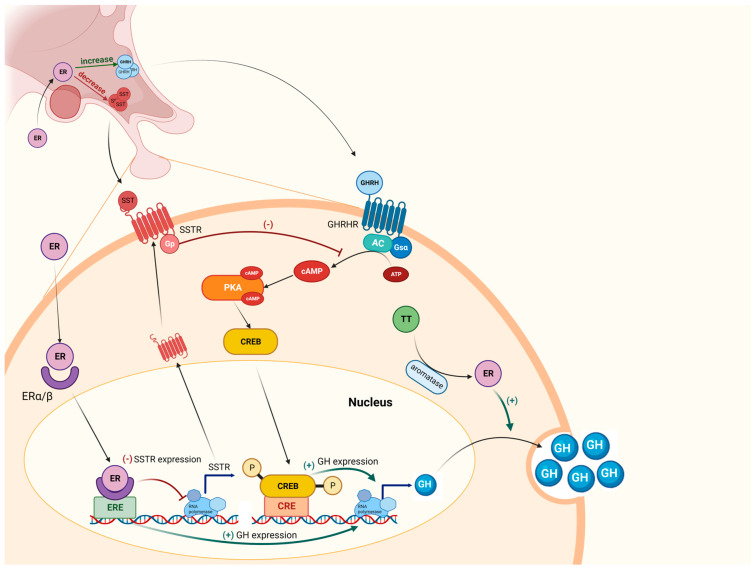
The GH and estrogen pathway in the somatotroph cell GH: growth hormone; ER: estrogen receptor type α or β; SST: somatostatin; SSTR: somatostatin receptor; cAMP: cyclic adenosine monophosphate; ATP: adenosine triphosphate; PKA: Protein kinase A; TT: total testosterone; CREB: cAMP response element binding protein; GHRH: GH releasing hormone; GHRHR: GHRH receptor; ERE: estrogen responsive elements; AC: adenylyl cyclase; Gsα: guanine nucleotide-binding protein Gs α subunit. Created with BioRender.com, accessed on 20 May 2023.

**Figure 2 ijms-24-09920-f002:**
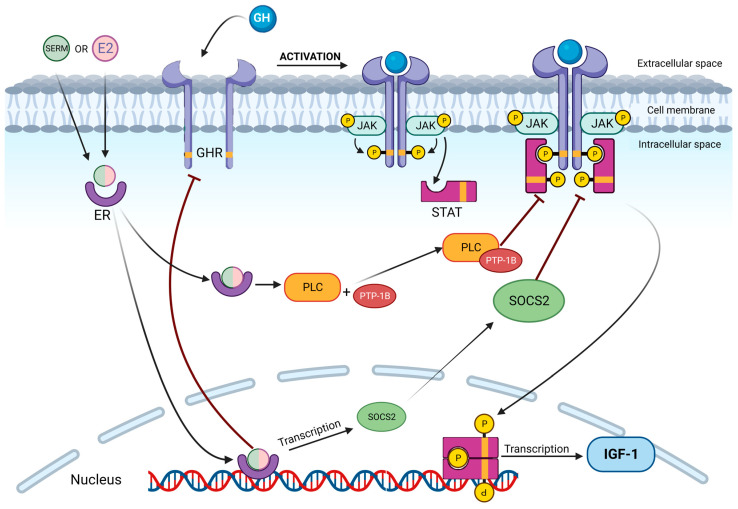
Estrogen effects on GH-responsive cell. GH binds to GHR, promoting its dimerization and the consequent activation of the JAK/STAT pathway, leading to the transcription of IGF1 and target genes. ER activation by estrogens or SERMs blunts IGF1 production through PLC and SOCS2 induction, which further blocks STAT phosphorylation. GH: growth hormone, IGF1: insulin-like growth factor, E2: estrogens, SERMS: selective estrogens receptor modulators, PLC: phospholipase C, JAK: janus kinase, STAT: signal transducers and activators of transcription, SOCS: suppressors of cytokine signaling, PTP-1B: protein tyrosine phosphatase-1B. Created with BioRender.com, accessed on 20 May 2023.

**Figure 3 ijms-24-09920-f003:**
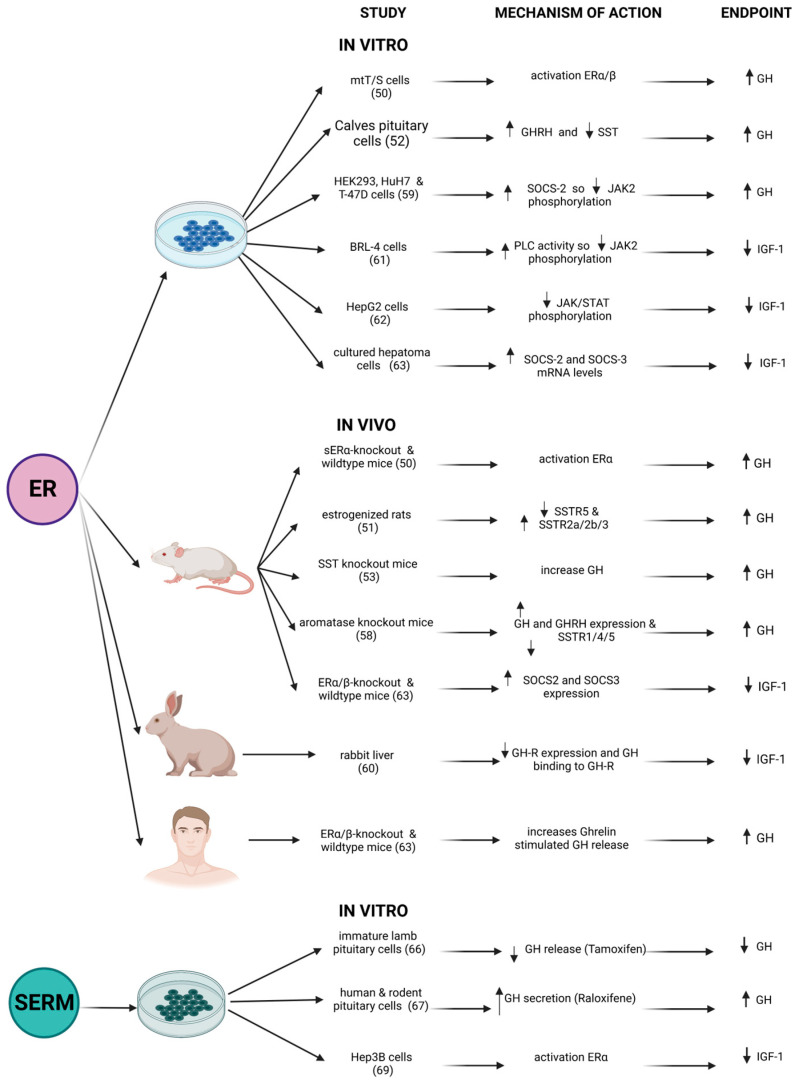
Summary of the studies targeting the estrogen receptors (ER) with estrogens or with selective estrogen receptor modulators (SERMs) in growth-hormone (GH) secreting cells, in animal models and in humans. IGF-1: insulin-like growth factor 1. ↑: Increased effect, ↓: reduced effect. The number of the reference (the same that appears in the manuscript) is depicted in brackets. Created with BioRender.com, accessed on 20 May 2023.

**Table 1 ijms-24-09920-t001:** Known effects of estrogens and SERMs versus GH and IGF1 levels and their mechanisms of action.

Sample	GH Effect	Mechanisms of Action	IGF1 Effect	Mechanisms of Action
Estrogens	↑	Increased Ghrelin sensivity	↓	Decrease GH receptor expression on target cells
		Increased GH mRNA production		PLC activation → inhibition of JAK/STAT signaling pathway
		Loss of negative feedback after IGF1 decrease		Upregulation of SOCS2 → impairment of JAK/STAT pathway
SERMs	↓	Anti-estrogenic effect at hypothalamus and pituitary level (in vitro) *	↓	Decrease GH receptor expression on target cells
				PLC activation → inhibition of JAK/STAT signaling pathway
				Upregulation of SOCS2 → impairment of JAK/STAT pathway

↑: Increased effect, ↓: reduced effect, GH: growth hormone, IGF1: insulin-like growth factor, PLC: phospholipase C, JAK: janus kinase, STAT: signal transducers and activators of transcription, SOCS: suppressors of cytokine signaling. * Increase of GH observed in vitro for raloxifene, not confirmed in vivo.

## Data Availability

All data presented are included in the manuscript.

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
