# Peer review of "Role of Estrogen and Estrogen Receptor in GH-Secreting Adenomas"

_ijms, 2023, doi:10.3390/ijms24129920_

Round 1
Reviewer 1 Report
Comment to the author:
In this review, the authors expound on the crucial role of estrogen and estrogen receptors in growth hormone adenomas. What’s more, the current state of research on the effects of estrogen and SERMs on GH/IGF1 was analyzed in this review, and concentrated on molecular pathways. These pathways have potential implications for the treatment of acromegaly. The subject matter of this paper is consistent with the scope of the International Journal of Molecular Sciences, and it is suggested that this manuscript may be considered for publication with major revisions, requiring attention to the following points:
1. The review is titled "Role of estrogen and estrogen receptor in somatotroph adenomas." However, the first part lacks sufficient information about growth hormone adenomas.
2. Figure 1, which outlines the pathway mechanisms of growth hormone and estrogen in growing trophoblast cells, does not provide enough detail. As a result, it is recommended to revise this section to include a more detailed and informative representation of the pathway mechanisms.
3. It is recommended to incorporate a schematic representation of the construction of the in vitro and vivo models in section three. This addition will enhance the paper's clarity, highlight the experimental design, and help readers understand the authors' findings in a more concrete manner.
4. In the fourth part, it is recommended to add a separate section on the application of SERMs in acromegaly.

The quality of the authors' English language is good in general. However, there are also some typos and grammatical errors need to be corrected. Please check it carefully.
For example:
(1) In the abstract moiety:
① line 1, please change increased to increasing
② line 4, please change first to firstly
③ line 11, please change the state of art to the state of the art
(2) In the key word moiety, the “estrogen receptor” should be “Estrogen receptor”
Please carefully check and correct the following typos and grammatical errors.
Author Response
Comments and Suggestions for Authors
Comment to the author:
In this review, the authors expound on the crucial role of estrogen and estrogen receptors in growth hormone adenomas. What’s more, the current state of research on the effects of estrogen and SERMs on GH/IGF1 was analyzed in this review and concentrated on molecular pathways. These pathways have potential implications for the treatment of acromegaly. The subject matter of this paper is consistent with the scope of the International Journal of Molecular Sciences, and it is suggested that this manuscript may be considered for publication with major revisions, requiring attention to the following points:
[Reply to reviewer 1]: We thank you for the positive review, which has instilled confidence in us and helped us revise the article.
- The review is titled "Role of estrogen and estrogen receptor in somatotroph adenomas." However, the first part lacks sufficient information about growth hormone adenomas.
[Reply to reviewer 1, question 1]: In the Introduction section we add more data regarding the pathogenesis of somatotroph adenomas.
- Figure 1, which outlines the pathway mechanisms of growth hormone and estrogen in growing trophoblast cells, does not provide enough detail. As a result, it is recommended to revise this section to include a more detailed and informative representation of the pathway mechanisms.
[Reply to reviewer 1, question 2]: We added further details in figure 1, which should help better understand and enlighten the molecular mechanisms explored in this review. However, we decided not to deepen further detailing, with the primary aim to focus on the molecular interplay between the GH pathway and estrogen pathway. In case a more detailed version of the picture is advisable, we are open to integrating further suggestions from the reviewers.
- It is recommended to incorporate a schematic representation of the construction of the in vitro and vivo models in section three. This addition will enhance the paper's clarity, highlight the experimental design, and help readers understand the authors' findings in a more concrete manner.
[Reply to reviewer 1, question 3]: To help in quickly visualizing the data collected from the in vivo and the in vitro models, we realized a graphical representation of the written paragraph, creating the figure 3.
- In the fourth part, it is recommended to add a separate section on the application of SERMs in acromegaly.
[Reply to reviewer 1, question 4]: section 4.2 of the manuscript is entirely dedicated to the clinical use of SERMs in patients with acromegaly. Integrating also the suggestions from the other reviewers, we added a table (reporting the clinical use of estrogens and SERMs) and a figure depicting the in-vitro and animal models. We deeply highlighted the crucial aspects of the use of SERMs, finally matching the purpose of the present review.
Comments on the Quality of English Language
The quality of the authors' English language is good in general. However, there are also some typos and grammatical errors need to be corrected. Please check it carefully.
For example:
(1) In the abstract moiety:
1 line 1, please change increased to increasing
2 line 4, please change first to firstly
3 line 11, please change the state of art to the state of the art
(2) In the key word moiety, the “estrogen receptor” should be “Estrogen receptor”
Please carefully check and correct the following typos and grammatical errors.
[Reply to reviewer 1, Comments on the Quality of English Language] we changed the manuscript accordingly.

Reviewer 2 Report
The review presented by Voltan et al. systematically addresses the role that estrogens and estrogen modulators may play in growth hormone signaling pathways.
The authors have submitted a well-written and presented manuscript, with figures and tables accompanying the text. Also, as a positive point, the authors have included a short paragraph about where and in what terms the search was performed. The review is of adequate length and includes the necessary references.
In the case of item "4.1. The use of oral estrogens in acromegaly" the authors have prepared a detailed table on the findings of their search. Also in this section and in "Effects of estrogens on GH-IGF1 axis: what in vitro and in vivo models show us" the authors have included two good figures to illustrate the biological processes.
This review suggests that the authors also include a table or summary figure in the sections that do not contain them. This would significantly improve the review and balance the figure/text ratio of the manuscript.
On the other hand, this reviewer also suggests that the authors correct the inappropriate use of abbreviations in the abstract. Abbreviations appear undefined and in the case of "SERMs" the abbreviation appears and then the term. The authors should revise the abstract since the rest of the manuscript, the use of abbreviations is correct.
Once again, I would like to congratulate the authors for their work and I hope that they will be able to implement the suggestions proposed for the publication of this review.
Author Response
Comments and Suggestions for Authors
The review presented by Voltan et al. systematically addresses the role that estrogens and estrogen modulators may play in growth hormone signaling pathways.
The authors have submitted a well-written and presented manuscript, with figures and tables accompanying the text. Also, as a positive point, the authors have included a short paragraph about where and in what terms the search was performed. The review is of adequate length and includes the necessary references.
In the case of item "4.1. The use of oral estrogens in acromegaly" the authors have prepared a detailed table on the findings of their search. Also in this section and in "Effects of estrogens on GH-IGF1 axis: what in vitro and in vivo models show us" the authors have included two good figures to illustrate the biological processes.
This review suggests that the authors also include a table or summary figure in the sections that do not contain them. This would significantly improve the review and balance the figure/text ratio of the manuscript.
On the other hand, this reviewer also suggests that the authors correct the inappropriate use of abbreviations in the abstract. Abbreviations appear undefined and in the case of "SERMs" the abbreviation appears and then the term. The authors should revise the abstract since the rest of the manuscript, the use of abbreviations is correct.
Once again, I would like to congratulate the authors for their work and I hope that they will be able to implement the suggestions proposed for the publication of this review.
[Reply to reviewer 2] We thank the reviewer for the helpful revision. We integrated the suggestions throughout the manuscript, as described below

Reviewer 3 Report
In the current article titled "Role of estrogen and estrogen receptor in somatotroph adenomas" by Giacomo Voltan et al, the authors have described acromegaly and the role of estrogen and SERMs in treating these patients. Through several studies described in this article, the authors have provided a holistic overview of the impact of estrogen and SERMs on regulating GH/IGF1, and how their levels could affect acromegaly treatment. While this study has good potential in the field and provides a lot of information, the following comments need to be addressed to further improve the quality of this article:
1. Title: Check the word 'receptor' -> 'receptors'.
2. Title: It would be appropriate to include the word 'Acromegaly' in the title of this article.
3. Line 29: Add comma '100,000' (or confirm 100 people?).
4. Line 30: Add comma '100,000' (or confirm 100 people?).
5. Line 39: What is OGTT? Elaborate on the abbreviations the first time it is mentioned.
6. Introduction: Any sex differences in the cases of acromegaly? Any sex differences in the treatment efficacy? Describe in a few lines.
7. Line 76: Provide a few examples of SERMs (you can list the same SERMs used in this article).
8. Introduction: Do women have reduced chances/cases of acromegaly? Describe in a few lines.
9. Figure 1: Describe 'ERE' in the figure legend.
10. Lines 178-179: The statement "The final...IGF1" seems to be incomplete. Please review and verify.
11. Figure 2: Describe 'PTP-1B', 'E2', and 'SERMs' in the figure legend.
12. Line 218: What are the relevant side effects observed due to treatment? State briefly.
13. Table 2: Describe 'SERMs' in the table legend.
14. Lines 231-240: How were IGF1 levels measured in these patients? Serum/plasma/blood test? Specify in these lines.
15. Line 243: Add period '56.8' (or confirm 56,8%?).
16. Lines 247-258: How often do IGF1 levels need to be monitored when on estrogen therapy? State briefly.
17. Lines 301, 302, 305, 310, 323, 334, 339: Confirm "mg/die" or "mg/day" (in English 'die' isn't used, so it would be appropriate to change the term to 'day').
18. Line 305: Add period '29.5' (or confirm 29,5%?).
19. Line 343: Check the word 'meta-analysis'.
20. Line 361: Correct the word 'lev-els' to 'levels'.
Please proofread the article for any incomplete sentences, typos, and errors.
Author Response
In the current article titled "Role of estrogen and estrogen receptor in somatotroph adenomas" by Giacomo Voltan et al, the authors have described acromegaly and the role of estrogen and SERMs in treating these patients. Through several studies described in this article, the authors have provided a holistic overview of the impact of estrogen and SERMs on regulating GH/IGF1, and how their levels could affect acromegaly treatment. While this study has good potential in the field and provides a lot of information, the following comments need to be addressed to further improve the quality of this article:
- Title: Check the word 'receptor' -> 'receptors'.
- Title: It would be appropriate to include the word 'Acromegaly' in the title of this article.
- Line 29: Add comma '100,000' (or confirm 100 people?).
- Line 30: Add comma '100,000' (or confirm 100 people?).
- Line 39: What is OGTT? Elaborate on the abbreviations the first time it is mentioned.
- Introduction: Any sex differences in the cases of acromegaly? Any sex differences in the treatment efficacy? Describe in a few lines.
- Line 76: Provide a few examples of SERMs (you can list the same SERMs used in this article).
- Introduction: Do women have reduced chances/cases of acromegaly? Describe in a few lines.
- Figure 1: Describe 'ERE' in the figure legend.
- Lines 178-179: The statement "The final...IGF1" seems to be incomplete. Please review and verify.
- Figure 2: Describe 'PTP-1B', 'E2', and 'SERMs' in the figure legend.
- Line 218: What are the relevant side effects observed due to treatment? State briefly.
- Table 2: Describe 'SERMs' in the table legend.
- Lines 231-240: How were IGF1 levels measured in these patients? Serum/plasma/blood test? Specify in these lines.
- Line 243: Add period '56.8' (or confirm 56,8%?).
- Lines 247-258: How often do IGF1 levels need to be monitored when on estrogen therapy? State briefly.
- Lines 301, 302, 305, 310, 323, 334, 339: Confirm "mg/die" or "mg/day" (in English 'die' isn't used, so it would be appropriate to change the term to 'day').
- Line 305: Add period '29.5' (or confirm 29,5%?).
- Line 343: Check the word 'meta-analysis'.
- Line 361: Correct the word 'lev-els' to 'levels'.
Please proofread the article for any incomplete sentences, typos, and errors.
[Reply to reviewer 3] We thank the reviewer for the helpful revision. We integrated the suggestions throughout the manuscript, as described above. More specifically, for point 2 we used GH-secreting adenoma (previous version somatotroph adenoma), because the term “acromegaly” is the clinical condition and most studies were animal or cellular models; for point 7, we added Table 3, providing estrogen and SERMs examples.

Reviewer 4 Report
Comments to Authors
The article “Role of estrogen and estrogen receptor in somatotroph adenomas” describes the impact of estrogen and SERMs on GH/IGF1 axis, focusing on molecular pathways and the possible implications for acromegaly treatment. It is proposed, that not only somatostatin receptor ligands, GH receptor antagonists, but also estrogens or SERMs could be used in the treatment of this disease.
There are some omissions. Some details and aspects are not written.
The main questions to the authors, that can expand the significance of the estrogen and estrogen receptors role in somatotroph adenomas are follows:
1.) How do the other steroid hormones act on GH/IGF1 production, for example, progestins or DHEA, or their antagonists
2.) Estrogen type, as well as SERM type should be characterized. What are the possible drug for acromegaly treatment? There is no any detailing about SERM - fulvestrant, tamoxifen, toremifen could be recommended
3.) What is the possible role of membrane ER could be in the GH/IGF1 axis regulation
There are some mistakes, omissions; some ABB are needed:
1.) Line 39: “paradoxical increase of GH during OGTT”: there is no ABB for OGTT
2.) 120 mg/die?
3.) contraceptive estro-progestin therapy?
4.) Line 83-90 s better to delete.
5.) Fig.1 have no comments or note in the text.

Author Response
Comments to Authors
The article “Role of estrogen and estrogen receptor in somatotroph adenomas” describes the impact of estrogen and SERMs on GH/IGF1 axis, focusing on molecular pathways and the possible implications for acromegaly treatment. It is proposed, that not only somatostatin receptor ligands, GH receptor antagonists, but also estrogens or SERMs could be used in the treatment of this disease.
There are some omissions. Some details and aspects are not written.
The main questions to the authors, that can expand the significance of the estrogen and estrogen receptors role in somatotroph adenomas are follows:
1.) How do the other steroid hormones act on GH/IGF1 production, for example, progestins or DHEA, or their antagonists
2.) Estrogen type, as well as SERM type should be characterized. What are the possible drug for acromegaly treatment? There is no any detailing about SERM - fulvestrant, tamoxifen, toremifen could be recommended.
3.) What is the possible role of membrane ER could be in the GH/IGF1 axis regulation
There are some mistakes, omissions; some ABB are needed:
1.) Line 39: “paradoxical increase of GH during OGTT”: there is no ABB for OGTT
2.) 120 mg/die?
3.) contraceptive estro-progestin therapy?
4.) Line 83-90 s better to delete.
5.) Fig.1 has no comments or note in the text.
[Reply to reviewer 4] We thank the reviewer for the helpful revision. We integrated all the suggested comments in the manuscript. In brief, we added a new table (Table 3) (also indicated by another Reviewer); the membrane estrogen receptor has not been studied yet in somatotroph cells or in situations of GH-IGF1 excess (further bibliographic research in Pubmed and Scopus was performed on 9th May 2023); the indication of bibliographic search methods has been appreciated by other 3 out of 5 reviewers, and therefore we decide to maintain it, giving the possibility to independently define the same state of the art on the present topic. Regarding Point 5 of the revision: Figure 1 was already mentioned in the first version of the manuscript and has its own Figure caption.

Reviewer 5 Report
Selective estrogen receptor modulators (SERMs) are synthetic, nonsteroidal agents that bind to the estrogen receptor, resulting estrogen-agonist or estrogen-antagonist type effects in different target tissues.
Differences in target tissue effects are related in part to unique SERM-receptor conformations and subsequent binding of different adaptor proteins. These proteins can facilitate or inhibit transcription of different DNA sequences, producing unique physiologic effects.
SERMs are now being used as a treatment for breast cancer, osteoporosis, postmenopausal symptoms and somatotroph adenomas.
In the past, estrogen was one of the first drugs used to treat acromegaly. Estrogens and SERMs treatments for acromegaly have received limited attention since the development of newer pharmacologic therapies.
SERMs represent an effective treatment to achieve control of IGF-1 levels in acromegalic women either as concomitant treatment for refractory disease, or where access to conventional therapy is restricted. Their use in men requires further study.
In the last years, the antagonists of growth hormone receptor became available, making possible IGF-1 control at the majority of patients with acromegaly.
This article presents data on the effectiveness of drugs and their place in the treatment algorithm of somatotroph adenomas.
The bibliography presented includes recent titles.
The article is well written and documented. I recommend it for publication.
Author Response
Selective estrogen receptor modulators (SERMs) are synthetic, nonsteroidal agents that bind to the estrogen receptor, resulting estrogen-agonist or estrogen-antagonist type effects in different target tissues.
Differences in target tissue effects are related in part to unique SERM-receptor conformations and subsequent binding of different adaptor proteins. These proteins can facilitate or inhibit transcription of different DNA sequences, producing unique physiologic effects.
SERMs are now being used as a treatment for breast cancer, osteoporosis, postmenopausal symptoms and somatotroph adenomas.
In the past, estrogen was one of the first drugs used to treat acromegaly. Estrogens and SERMs treatments for acromegaly have received limited attention since the development of newer pharmacologic therapies.
SERMs represent an effective treatment to achieve control of IGF-1 levels in acromegalic women either as concomitant treatment for refractory disease, or where access to conventional therapy is restricted. Their use in men requires further study.
In the last years, the antagonists of growth hormone receptor became available, making possible IGF-1 control at the majority of patients with acromegaly.
This article presents data on the effectiveness of drugs and their place in the treatment algorithm of somatotroph adenomas.
The bibliography presented includes recent titles.
The article is well written and documented. I recommend it for publication.
[Reply to reviewer 5] We thank the reviewer for the positive comments.

Round 2
Reviewer 4 Report
All the comments are corrected.
The article have been significantly improved.
There is one moment: it is better to correct estro-progestin pills to estrogene-progestin patients pills.